# Real-time telemetry monitoring of oxygen in the central complex of freely-walking *Gromphadorhina portentosa*

**Pier Andrea Serra**[1,2,3]*, **Paola Arrigo**[1], **Andrea Bacciu**[1], **Daniele Zuncheddu**[1], **Riccardo Deliperi**[1], **Diego Antón Viana**[1], **Patrizia Monti**[1], **Maria Vittoria Varoni**[4], **Maria Alessandra Sotgiu**[5], **Pasquale Bandiera**[5], **Gaia Rocchitta**[1,3]

1 Department of Medical, Surgical and Experimental Medicine, Medical School, University of Sassari, Sassari, Italy, 2 Institute of Sciences of Food Production, Italian National Research Council, Sassari, Italy, 3 Mediterranean Center for Disease Control, University of Sassari, Sassari, Italy, 4 Department of Veterinary Medicine, Medical School, University of Sassari, Sassari, Italy, 5 Department of Biomedical Sciences, Medical School, University of Sassari, Sassari, Italy

* paserra@uniss.it

**Data Availability Statement:** All relevant data are within the manuscript and its Supporting Information files.

## Abstract

A new telemetric system for the electrochemical monitoring of dissolved oxygen is showed. The device, connected with two amperometric sensors, has been successfully applied to the wireless detection of the extracellular oxygen in the central complex of freely-walking *Gromphadorhina portentosa*. The unit was composed of a potentiostat, a two-channel sensor conditioning circuit, a microprocessor module, and a wireless serial transceiver. The amperometric signals were digitalized and sent to a notebook using a 2.4 GHz transceiver while a serial-to-USB converter was connected to a second transceiver for completing the communication bridge. The software, running on the laptop, allowed to save and graph the oxygen signals. The electronics showed excellent stability and the acquired data was linear in a range comprised between 0 and -165 nA, covering the entire range of oxygen concentrations. A series of experiments were performed to explore the dynamics of dissolved oxygen by exposing the animals to different gases (nitrogen, oxygen and carbon dioxide), to low temperature and anesthetic agents (chloroform and triethylamine). The resulting data are in agreement with previous $O_2$ changes recorded in the brain of awake rats and mice. The proposed system, based on simple and inexpensive components, can constitute a new experimental model for the exploration of central complex neurochemistry and it can also work with oxidizing sensors and amperometric biosensors.

## Introduction

Synaptic transmission requires a large amount of energy that is satisfied through the use of different energy substrates [1,2] but it is through the mitochondria that a higher production of ATP can be obtained [3,4]. Oxygen represents the pivotal oxidative substrate for neurochemical reactions in which energy is produced. The levels of $O_2$ dissolved in the tissues depends on

**Funding:** PAS: The research was supported by the University of Sassari ("Fondo di Ateneo per la Ricerca 2019"); GR: The research was supported by the University of Sassari ("Fondo di Ateneo per la Ricerca 2019"); www.uniss.it. The funders had no role in study design, data collection and analysis, decision to publish, or preparation of the manuscript.

**Competing interests:** The authors have declared that no competing interests exist.

the compensation between provision and local utilization [5] and it participates in neural metabolism of glucose [6] or lactate [7] used for ATP production. In this regard, recent findings demonstrated that insects not only show a metabolic cross-talk between glial and neurons that closely resembles that observed in neural metabolism of vertebrates, but also it seems to be present a direct link between brain metabolic dynamics and behavioral phenotypes [8].

In order to monitor the changes of extracellular oxygen levels, some amperometric sensors have been employed, which were previously implanted in murine animal models, as already disclosed in our previous researches [9,10]. For a more efficient reduction of $O_2$ to water, carbon-based sensors have been preferred to platinum-based ones, also because of their invariability and minor contamination of the surface [9,11].

The detection of $O_2$ occurs by means of an electrochemical reduction when a cathodic potential is applied to the sensor surface. The reaction could take place by means of several steps [12]:

$$O_2 + 2H^+ + 2e^- \rightarrow 2O_2 \tag{1}$$

$$H_2O_2 + 2H^+ + 2e^- \rightarrow H_2O \tag{2}$$

or in a single passage [12], as follows:

$$O_2 + 4H^+ + 4e^- \rightarrow H_2O \tag{3}$$

Recently, neurochemicals have been monitored in real time by means of several telemetry systems [13,14] and some of them were specially conceived for monitoring only oxygen in awake rodens [9,10]. In this research we display a miniaturized telemetric system, proposed as the evolution of previous designs and successfully used for detecting dissolved oxygen in central complex (CX) of *Gromphadorhina portentosa*.

The insect CX is a core structure of the insect brain: it consists of a midline conglomerate of four brain areas that is conserved across all insects [15]. The role of the CX has been studied for years revealing several functions such as motor control, navigation, sensory integration, attention, control of sleep and memory [15].

In this study, the stereotaxic procedure of implantation of neurochemical sensors in the CX of anaesthetized insects and the telemetric monitoring of dissolved oxygen in awake, freely-walking animals, has been achieved for the first time. Several experiments have been conducted to show the capability of the system to measure changes in oxygen levels in the CX extracellular space with a sub-second time resolution.

The presented system can be used as a fast and valid model for investigating the neurochemical response of insect neural cells to physico-chemical stimulations, or to the administration of different drugs, in terms of oxygen consumption.

An important assessment must also be made on the ethical level on the possibility of reducing the pain perceived by the animals used for the experimentation and, where possible, replace the model [16]. Even if the issue is still controversial [17,18], according to the current state of knowledge, it would seem that only vertebrates perceive pain [19].

In light of these reasonings and on the basis of the results obtained in this study, although the proposed experimental model can be considered more distant from the human brain than that of vertebrates (i.e. zebrafish or rodents), it could allow to answer well-posed scientific questions by replacing or reducing the experimental use of vertebrates [16], mammals in particular.

## Materials and methods

### Reagents and solutions

All analytical grade chemicals were employed as provided and solubilized in MilliQ water. Chloroform (Supelco, ref. 1.02442), triethylamine (Sigma, ref. 471283), graphite powder (Fluka, ref. 78391) and collodion solution (4% cellulose nitrate in ethanol/diethyl ether from Fluka, ref. 09986) were bought from Sigma-Aldrich (Milano, Italy). The phosphate buffer saline (PBS) solution was obtained with the following concentrations, expressed in mM: NaCl (137), $KH_2PO_4$ (1.47) KCl (2.7), and $Na_2HPO_4$ (8.1) and then buffered to pH 7.4. All the compounds for the PBS solution were from Sigma-Aldrich (Milano, Italy). Nitrogen, ultrapure (>99.9%) oxygen and carbon dioxide were acquired from Sapio s.r.l Special Gases Division (Caponago, Italy). $O_2$- and $N_2$-saturated solutions for sensors calibrations, were obtained by bubbling 10 mL of PBS with the relative gas for 1 h and used right away. The oxygen *in-vitro* calibrations were performed at room temperature (25°C) and pressure (1 atm) prior to the *in vivo* implantation.

### Animals

Adult male cockroaches (n = 16), bred in our laboratory colony and weighing 7.5–8.6 g, were used for *in vivo* experiments. The animals were maintained in plastic cages (42x21x30 cm) under controlled conditions of temperature and humidity (12 h/12 h inverted light/dark cycle, light at 19.00 p.m.; temp: 26–28°C; hum: >40%) with food (Purina Dog Chow) *ad libitum*. Several inverted cardboard flats were used as shelter and wood-shaving as bedding; the cages were cleaned once a week [20]. Water was administered through a 5% starch gel and every 7–10 days fresh fruit (apple, pear and banana) was introduced as dietary supplement. Prior to the start of any experimental procedure, the apparent good health of the selected cockroach was assessed by observing the animal behavior for 30 min. All procedures and experiments were performed between 10.00 a.m. and 16.00 p.m.

### Preparation and calibration of oxygen sensors

The $O_2$ sensors (Fig 1) were made by slightly varying an already described procedure [9] making use of insulated multicore copper wires of 30 mm in length (single core o.d. Ø = 150 μm). After the removal of the insulation (4 mm), a single core wire was introduced for 4 mm in a silica capillary tube of 5 mm in length (i.d. Ø = 180 μm, Polymicro Technologies, Phoenix, AZ). A carbon-composite disk of 180 μm of diameter with an area of $2.5 \times 10^{-4}$ cm$^2$ was obtained by blending 850 mg of graphite with 500 mg of Araldite-M (Sigma-Aldrich, Milan, Italy) and then adding the mixture with 200 mg of hardener. After that, the silica capillary tubing was filled with the composite until reaching the copper wire. After 30 min, the insulation was slid so as to free only the apical 2 mm of the capillary and then fixed with no-conducting epoxy resin. The working electrode (WE) was left at 40°C for 24 h and then smoothed by means of a high speed drill (Dremel 300) provided with an aluminum oxide grinding wheel. Cellulose nitrate layering was done by dipping the WE in the collodion solution for three times and letting it dry up at 40°C for 60 min, after every coating. $O_2$ reduction was investigated on by means of cyclic voltammetry (CV37 voltammograph, BAS, Bioanalytical Systems Inc., West Lafayette), applying a potential range starting from -350 mV vs Ag/AgCl pseudo-reference electrode (RE), with a scan rate of 25 mV/s [9]. For *in vitro* and *in vivo* calibrations, as well as experiments, constant potential amperometry (CPA) was employed, by applying a fixed potential of -400 mV vs Ag/AgCl. No remarkable interference signals were monitored during exposition of the sensors to different electroactive compounds such as uric acid (UA) and ascorbic

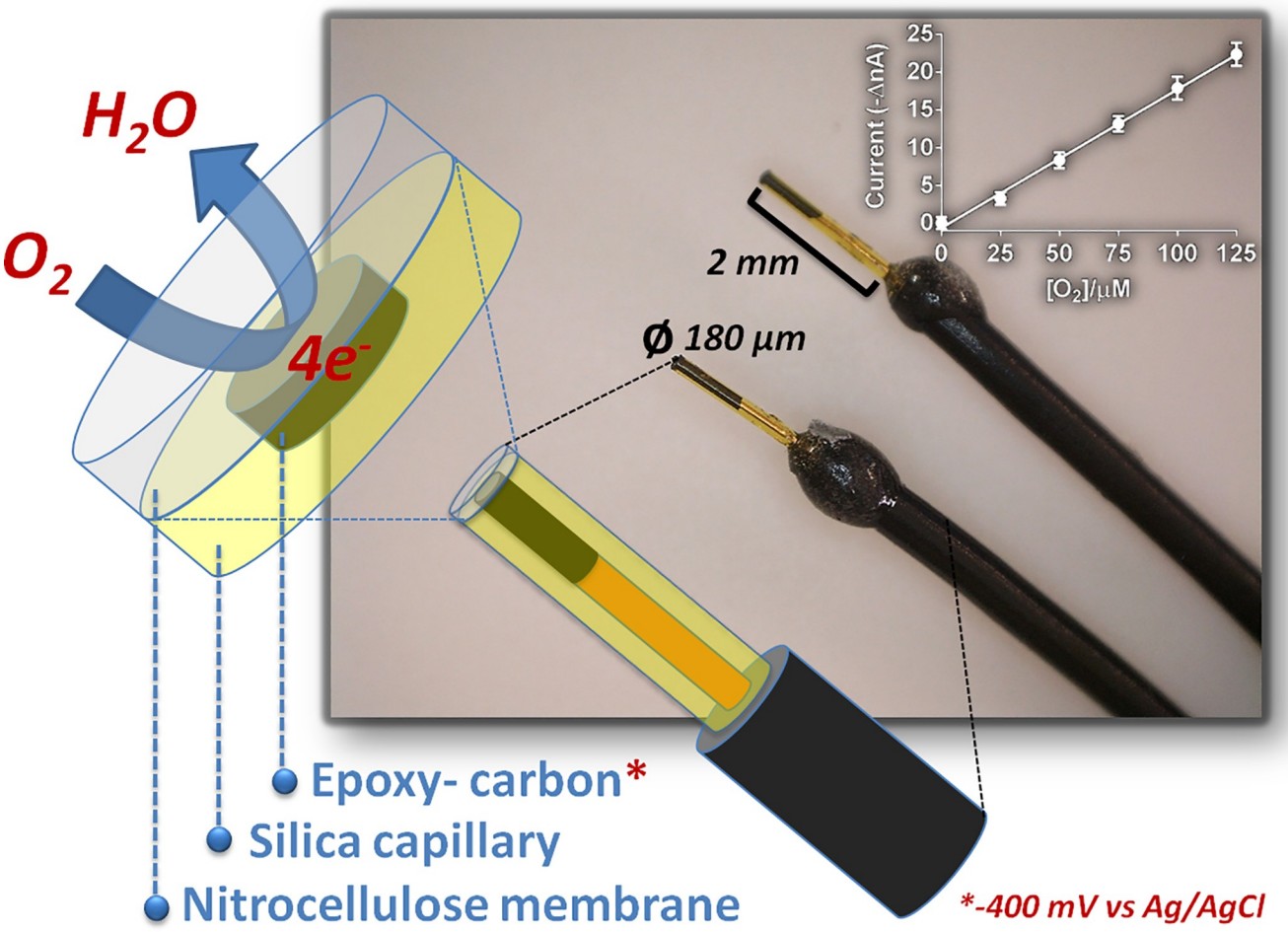

**Fig 1. Oxygen sensors design and calibrations.** Disk sensors were made by using insulated copper wires inserted into silica capillaries, connected with epoxy-carbon and then covered with collodion membrane (nitrocellulose). The $O_2$ calibrations (upper-left inset) were performed in $N_2$-purged PBS by applying a negative potential of -400 mV vs Ag/AgCl RE.

acid (AA), or dopamine (DA), octopamine (OT) and 5-hydroxy-tryptamine (5-HT), molecules potentially present in the extracellular compartment of the CX, even at pharmacologically considerable concentrations (0.25 mM for UA and AA; up to 1 μM for the other neurochemicals).

A precise calibration was carried out at low concentrations of $O_2$ (Fig 1, upper-left inset) after having connected the microsensors to the telemetric device (see the dedicated paragraph) and adding up, to a 10 milliliters of $N_2$-purged PBS, defined volumes of a 100% $O_2$ solution (+200, +204, +208, +212, and +216 μL). The auxiliary (AE) and the RE electrodes were built by uncovering 1 mm of a teflon-insulated silver wire of 30 mm in length (o.d. Ø = 250 μm, Advent Research Materials, Suffolk, U.K.): in particular, the RE electrode was obtained by a further modification (for obtaining a not soluble layer of AgCl) by dipping it in a saturated KCl solution and by applying a potential of +500 mV for 1 min.

### Stereotaxic surgery

Stereotaxic surgery was performed under $CO_2$ anesthesia (90 s exposure in a sealed container). The animal was immobilized in a stereotaxic frame adapter (a detailed description is provided in S1 File) to allow the precision procedures necessary for the measurement of stereotaxic

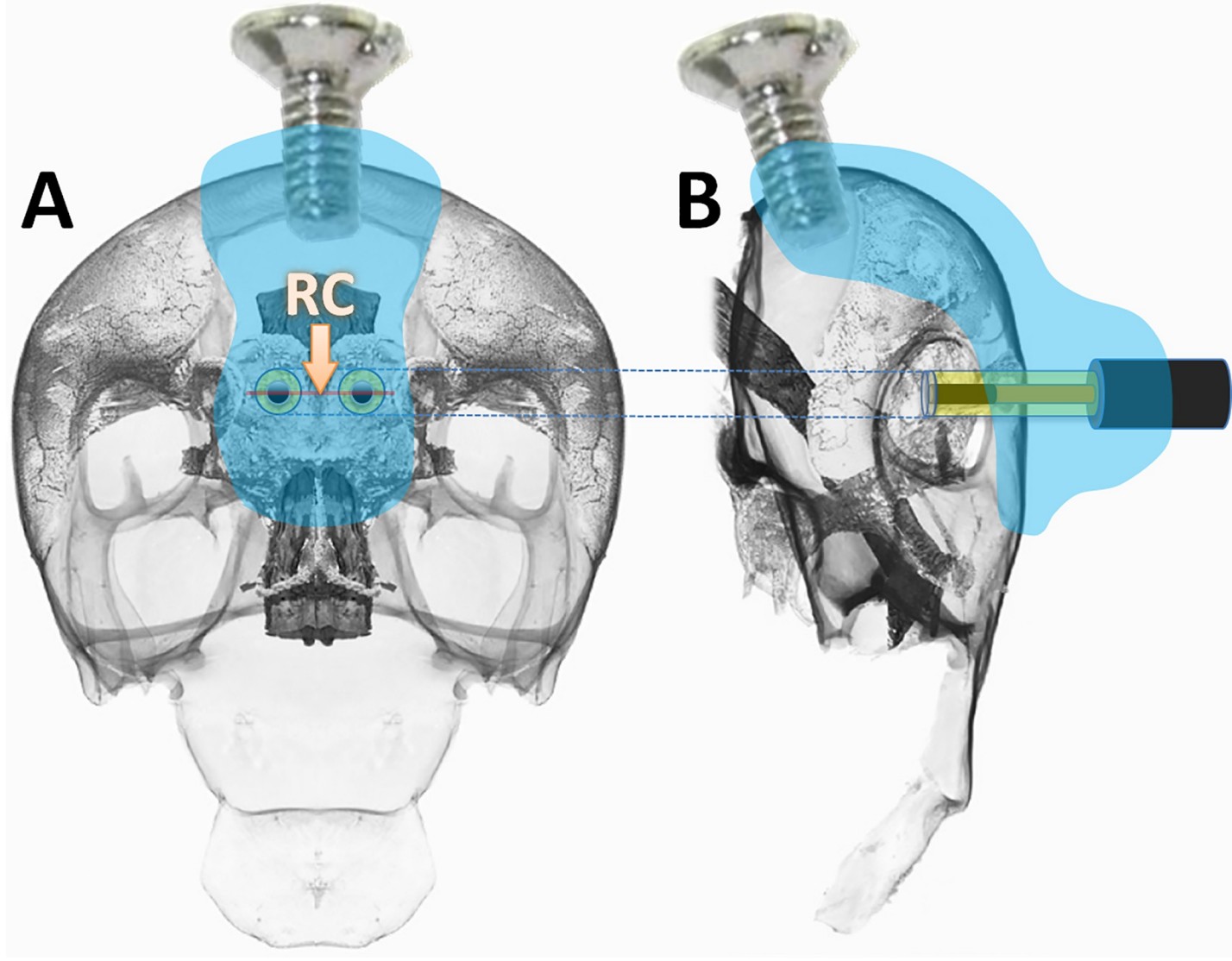

**Fig 2. Stereotaxic surgery, implantation and fixing of the O$_2$ sensors.** The O$_2$ sensors were implanted in the right and left sides of the CX by using specific coordinates calculated starting from the reference coordinate (RC, see text). The sensors were lowered for 1 mm and fixed by using a screw and UV-curable dental cement (light blue drawing area). A: frontal view; B: lateral view.

coordinates, the drilling of the exoskeleton and the insertion of the sensors. The adapter was then fixed to a Kopf sterotaxic frame (Kopf model 963 ultra-precise small animal stereotaxic, Tujunga CA, USA) by means of the incisive and auricular bars.

Since an anatomical atlas of the *Gromphadorhina portentosa* head is not available yet, the organization of the cephalic compartment of the *Periplaneta americana* has been taken as a reference [21], then the stereotactic coordinates were refined according to the current organization of the insect brain [22]. The reference coordinate (RC) has been identified as the median point on the horizontal line joining the two ocellar spots. The oxygen sensors (WEs) were implanted in the right and left sides of the CX (Fig 2) using the following sterotaxic coordinates: 0.0 A/P, ±1.2 M/L from RC and -1.0 D/V from the exoskeleton surface. A small screw was inserted for 1 mm in the midline of the rostral part of the head and joined to the WEs with UV-curable dental cement (Flow-It™ ALC™, Pentron-Danaher Corp., Washington D.C., USA).

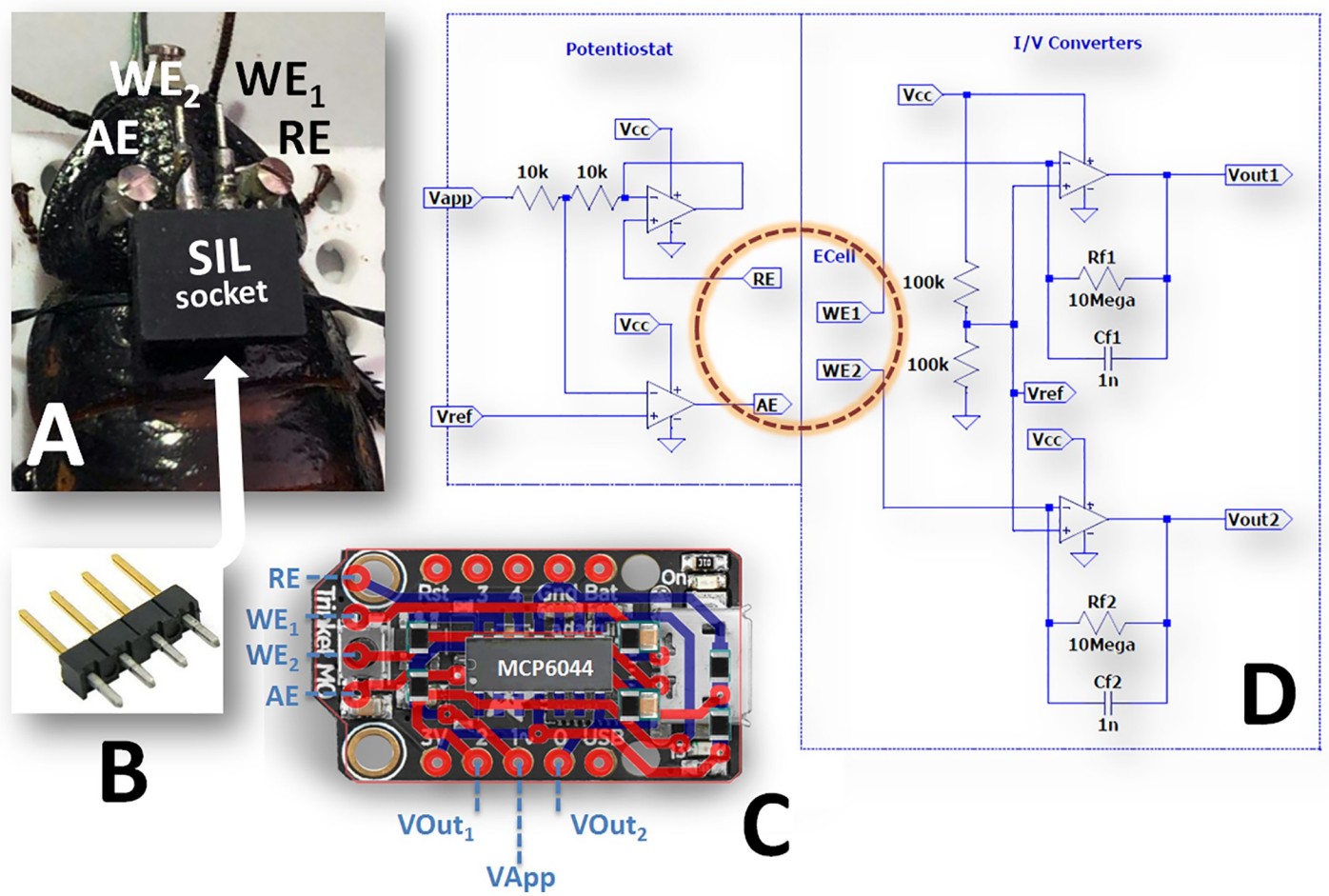

**Fig 3. Electrodes holder and telemetry system.** The 4-pin SIL (female) socket allowed the connection with RE, AE and WEs. The two screws inserted in the pronotum reinforced the telemetric device holder adhesion (A). Four SIL male pins (B) established a mechanical and electrical bridge between the electrodes and the telemetry device (C). The electronic circuits of the potentiostat and the current-to-voltage converters are represented in panel D (for the detailed description see S1 File).

Some steps of the neurosurgical procedure are illustrated in S1 File. The RE and AE electrodes already soldered to a single-in-line (SIL) 4-pin socket were implanted in the "horns" of the pronotum, and fixed with two screws to reinforce the telemetric device holder adhesion (Fig 3A). The WEs were then connected to the SIL (female) socket which was secured to the pronotum using dental cement (Paladur, Heraeus Kulzer, GmbH). Four SIL male pins (Fig 3B) guaranteed the mechanical and electrical connections between the electrodes and the telemetric device (Fig 3C). Following the surgery, the animals were placed in plastic cages (24x16x20 cm) and preserved in a temperature-, humidity- and light-controlled habitat. Animals had free access to food and water.

## Telemetry system

The electronic components for the construction of the potentiostat and the current-to-voltage converters (Fig 3D) were from Farnell-In-One s.p.a. (Milano, Italy). The 2.4 GHz wireless serial port modules (JDY-40) with integrated antennas were from Shenzhen Xintai Micro Technology Co. Ltd. (Shenzhen, China) while the serial-to-USB-convert was made around the CP2102 integrated circuit (Silicon Laboratories Inc., Austin TX, USA). The 10-bit digital-to-

analog (DAC) and the 12-bit analog-to-digital (ADC) converters were an integral part of the microcontroller unit (MCU) selected for this project (SAMD21 inside a Trinket M0 board Adafruit, New York NY, USA). The amperometric section of the telemetric device was built using a quad single-supply operational amplifier MCP6044 (Arizona Microchip, Chandler AZ, USA), multilayer ceramic chip capacitors (MLCC) and precision metal oxide thick film resistors (250 mW, 0.1% tolerance, Ohmite, Rolling Meadows, IL). The surface mount devices (SMD) were soldered on a double side printed circuit board (PCB) and a 180 mAh lithium-ion-polymer (Li-Poly) battery completed the design. Each part of the electronics was Pb-free and compliant to RoHS directives. The weight of the device was 8.7 g (plus 5.3 grams of battery). A detailed description of the hardware, firmware and software is provided in S1 File.

### *In vivo* experimental procedures

The polarization of $O_2$ microsensors occurred 24 h after surgery (day 1) subsequently to the connection of the telemetric device with its battery to the SIL socket: this arrangement allowed the animal free walking (see S1 Movie). When a stable $O_2$ baseline was reached, mild hyperoxia and hypoxia were produced on day 1 by exposing the cockroach to $N_2$ and $O_2$ for 1 min, respectively in an open chamber. Prolonged exposure to pure oxygen (5 min) and anesthetics administration were performed in a sealed chamber 48 h after neurosurgery (day 2). $CO_2$ and low temperature experiments were conducted within the days 3 and 6 after the stereotaxic surgery. All experiments with gases and anesthetics were carried out in plastic boxes (24x16x20 cm). For low temperature experiments, a polystyrene box (36x18x24 cm) was partially filled with freezer packs and the temperature was constantly monitored with a precision thermometer; the animal was then introduced in the box when the temperature reached a stable value around -2°C.

### Statistical analysis

All graphs and statistical analysis were performed using GraphPad Prism version 5.03 for Windows (GraphPad Software, La Jolla California USA, www.graphpad.com). Concentrations of dissolved $O_2$ were given in micromolar (μM) while oxygen currents derived from the molecule reduction were defined in nanoamperes (nA) or given as baseline-subtracted raw data (ΔnA). In order to ameliorate the comprehension of data, the sign of the oxygen currents was inverted. Following the *in-vitro* calibrations, a graph, plotting currents versus oxygen concentration, was obtained and the linear regression was calculated. The modification in $O_2$ levels were evaluated as absolute variations respect to the corresponding baseline, while the oxygen concentrations were calculated, and then expressed in μM, using pre-implantation slopes resulting from linear regressions. The statistical significance of $O_2$ variations was determined by employing paired t tests between the means (± standard deviation, SD) of successive measurements before (baseline) and during the highest extent of oxygen changes as a result of the experimental treatments. When ventilatory abdominal movements (VAM) were observed (in particular after CO2 or anesthetics exposure), the corresponding $O_2$ currents were compared to the baseline and the maximum oxygen excursion consequent to the treatment.

## Results

### Electronics calibration and sensors response to oxygen

As illustrated in previous papers [9,10,13,14], a dual-channel sensor current simulator (current source) was used in order to validate the analog electronics of the telemetric device. The sensor current simulator was made by connecting two 10 MΩ resistors to the WEs and RE/AE inputs

of the telemetric device. By fixing Vapp to -0.4 V, a current of -40nA was generated and converted to voltages (Vout1 and Vout2) by the transimpedance amplifiers as showed in the Fig 3. The electronics tests were made in standard laboratory conditions with a distance between the telemetric unit and the PC of about 5 m. The Vout signals were recorded for 4 consecutive hours of continuous operation and a maximum deviation of 5 mV was monitored while the electronic noise (unconnected sensor current simulator) was around 15 pA. The power consumption was calculated by connecting a digital milliammeter in series with the Li-Poly battery and resulted in 32 mA with a verified operation time of more than 5 h.

Before the neurosurgical implantation, the sensors were calibrated by adding known volumes of a $O_2$-saturated PBS solution (1.25 mM) to nitrogen-purged PBS. In detail, the oxygen sensors were connected to the telemetric device and placed in a beaker containing 10 mL of nitrogen-purged PBS; Vapp was then applied and a stable baseline was reached (14 ± 2 nA). The $O_2$-saturated PBS was added and the beaker content was quickly stirred after each injection. The oxygen signals (Vout1 and Vout2) were taken under quiescent conditions. The calibration (Fig 1, upper-left inset) displayed excellent linearity with a slope of 0.183 ± 0.014 nA $\mu M^{-1}$ of $O_2$ ($r^2$ = 0.997). The response time of sensors was <1 s.

## Baseline levels of $O_2$ in the CX extracellular space

At day 1, baseline CX oxygen currents were monitored after the stabilization of the sensors, which were achieved after an interval of 32 ± 14 min (n = 8) from the activation of the system. Before experiments, an interval of 15 minutes of raw data (comprising 4500 consecutive measurements of currents) was recorded. Currents were then averaged and expressed as mean ± SD. From recordings, $O_2$ baseline was calculated as 24 ± 5 nA (Fig 4A). Taking into account that the background current of the sensor, when measured in nitrogen-saturated PBS, was found to be around 14 nA, it was possible to extrapolate the $O_2$ concentration from pre-calibrations data; the baseline $O_2$ level was assessed to 55 ± 27 μM.

## Effects of $O_2$ and $N_2$ exposure on CX oxygen currents

The administration of pure gases (oxygen and nitrogen) to the cockroaches, when placed in their own (open) boxes, occurred by the connection of gas cylinders to the box and making them flow alternatively at high fluxes (1 L/min). With this procedure, animals were obliged to breath a mixture of air-$N_2$ or air-$O_2$ for 1 min, as shown in Fig 4B. Between $N_2$ and $O_2$ administration, the box was aerated with a fan in order to allow the rapid reintroduction of air. Following the pure $N_2$ exposure, the $O_2$ signals were significantly lowered by -6.3 ± 0.6 nA (-34.4 ± 3 μM; p<0.01 vs baseline). They got back to the baseline at the end of the $N_2$ administration. On the contrary, the pure oxygen administration produced a short-lasting increase (+-15.6 ± 1.1 nA corresponding to +85.2 ± 6 μM) in the $O_2$ currents (p<0.01 vs baseline and p<0.001 vs $N_2$ exposure). In a separate series of experiments, the housing bowl was sealed with a plexiglass lid and pure oxygen was introduced for 5 min (Fig 4C). As result of this treatment, a sustained increase of the oxygen signals (+47.2 ± 7 nA corresponding to +258.9 ± 38 μM; p<0.001 vs baseline) was observed with the return to baseline (+3.8 ± 3.4 nA) in 34 ± 12 min.

## Effects of $CO_2$ and low temperature exposure on CX oxygen currents

$CO_2$ administration for 90 s in a sealed container (Fig 5A) induced a rapid decrease in the oxygen signal -8.3 ± 1.9 nA (-45.4 ± 10.3 μM; p<0.001 vs baseline) for a duration of 6.6 ± 2 min followed by a rapid ascent up to levels higher than baseline +2.4 ± 1.1 nA (+13.1 ± 6 μM; p<0.05 vs baseline). During this mild hyperoxic phase, ventilatory abdominal movements (VAM) were observed in all animals. Around 30 min after the carbon dioxide exposure, the

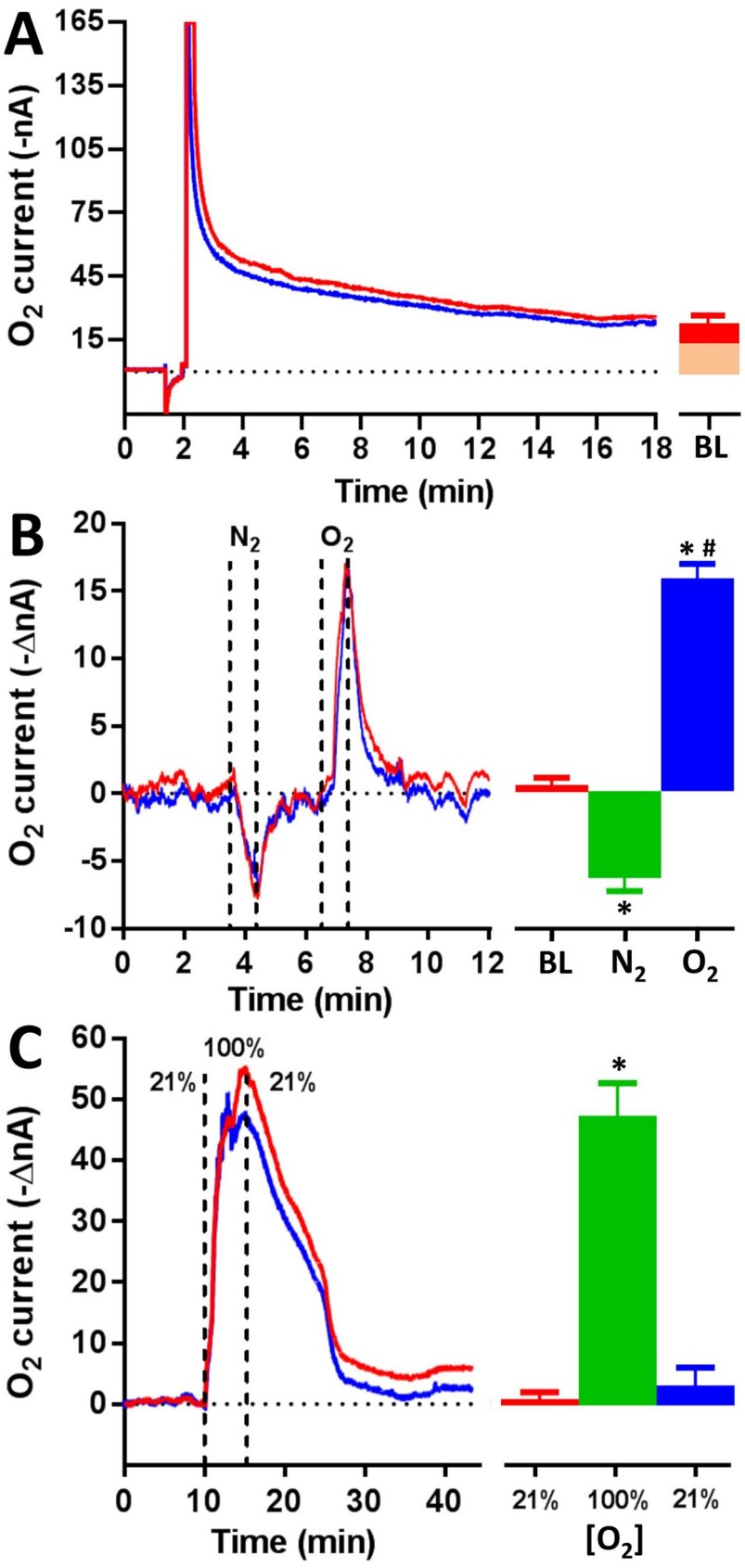

**Fig 4. *In vivo* settling down and effects of N$_2$ and O$_2$ exposure on oxygen currents.** The initial stabilization of sensors currents was used for calculating the O$_2$ baseline (A). Only one portion of the signal represents the oxygen current (the red-colored area of the column in A) while the remaining portion (orange-colored area) is the averaged background current of the sensors (see text). The short exposure (1 min) of the cockroach to pure nitrogen (B), determined a significant decrease in O$_2$ currents (* = p <0.01 vs baseline), that returned to baseline values immediately after N$_2$ removal. Conversely, the rapid exposure to pure oxygen (B), caused a short-lasting rise in the oxygen current (* = p<0.01 vs baseline; # = p <0.001 vs N$_2$ exposure). The prolonged exposure (5 min) to pure oxygen, in a sealed chamber, induced a sustained increase (C) increase of the oxygen currents (* = p <0.001 vs baseline) in about half an hour (# = p <0.001 vs O$_2$ exposure). BL = baseline.

cockroaches completely recovered from anesthesia and the oxygen currents came back to baseline values. When the animals were exposed to low temperatures (~-2˚C for 30 min), the oxygen changes illustrated in Fig 5B were observed (-6.2 ± 1 nA; -33.8 ± 5.5 μM; p<0.001 vs baseline). After the animals were brought back to their cages (at room temperature), oxygen currents stabilized at levels significantly lower than the basal ones (-4.1 ± 0.6 nA; -22.4 ± 3.3 μM; p<0.05 vs low temperature) and remained so for 4–7 hours. No VAM were observed.

## Effects of chloroform and triethylamine administration on CX oxygen currents

Chloroform (CHCl$_3$) and triethylamine (C$_6$H$_{15}$N or Et$_3$N) were administered by injecting 3 mL of the anesthetic directly into a cotton ball inside a box which was sealed immediately after. When the air was saturated (after15 min) the animal was introduced through a small aperture and left inside for 1 minute. Chloroform exposure (Fig 6A) resulted in a rapid decrease of oxygen currents (-7.1 ± 1.7 nA; -38.8 ± 9.2 μM; p<0.001 vs baseline) for a duration variable (from 5 to 23 min) followed by a gradual increase up to baseline levels (+0.7 ± 3.6 nA; +3.8 ± 19.6 μM; p>0.05 vs baseline). The return to baseline takes place slowly with a not always predictable timing. VAM were observed only in two animals. The administration of triethylamine (Fig 6B) induced a dramatic O$_2$ decrease (-9.4 ± 2.6 nA; -51.4 ± 14.2 μM; p<0.001 vs baseline) with a return to baseline in 17 ± 8 min. VAM were observed in most cockroaches and the averaged oxygen currents tended to exceed basal values (+2.7 ± 1.8 nA; +14.8 ± 9.8 μM; p>0.05 vs baseline). The anesthetic onset time of both drugs was inconstant and some animals showed hypomotility up to 12 hours.

## Discussion

### Operation of the telemetric device

The analog circuitry of the telemetric device was designed for operating at 3.3V. While two operational amplifier (OPAs) were dedicated to the potentiostatic circuit, two extra OPAs were used for the current-to-voltage (I/V) converters without further hardware processing of the analog signals (Fig 3D). The design was directly derived from previous studies [9,10,13,14] with two main differences: the first related to the potentiostat design and the second linked to the absence of the zener diode then used as voltage reference (Vref). In the current design, Vref has been fixed to 1.65V through a resistive divider directly connected to the 3.3V precision voltage stabilizer of the Trinket M0 board. In this manner we successfully overcame the limitations of the previous projects [9,10], in which the system could be used alternatively in reduction or oxidation mode [10]. In the present study, we setup the device for allowing the reduction of the oxygen on carbon surface by applying a potential of -400 mV vs Vref (Vapp = 1.25V); however, in future researches, it will be possible to operate in oxidation mode with Vapp > Vref and using the system with oxidizing sensors or amperometric biosensors (i.e. oxidase-based biosensors for glucose or lactate detection [23]). As shown in the Fig 3D,

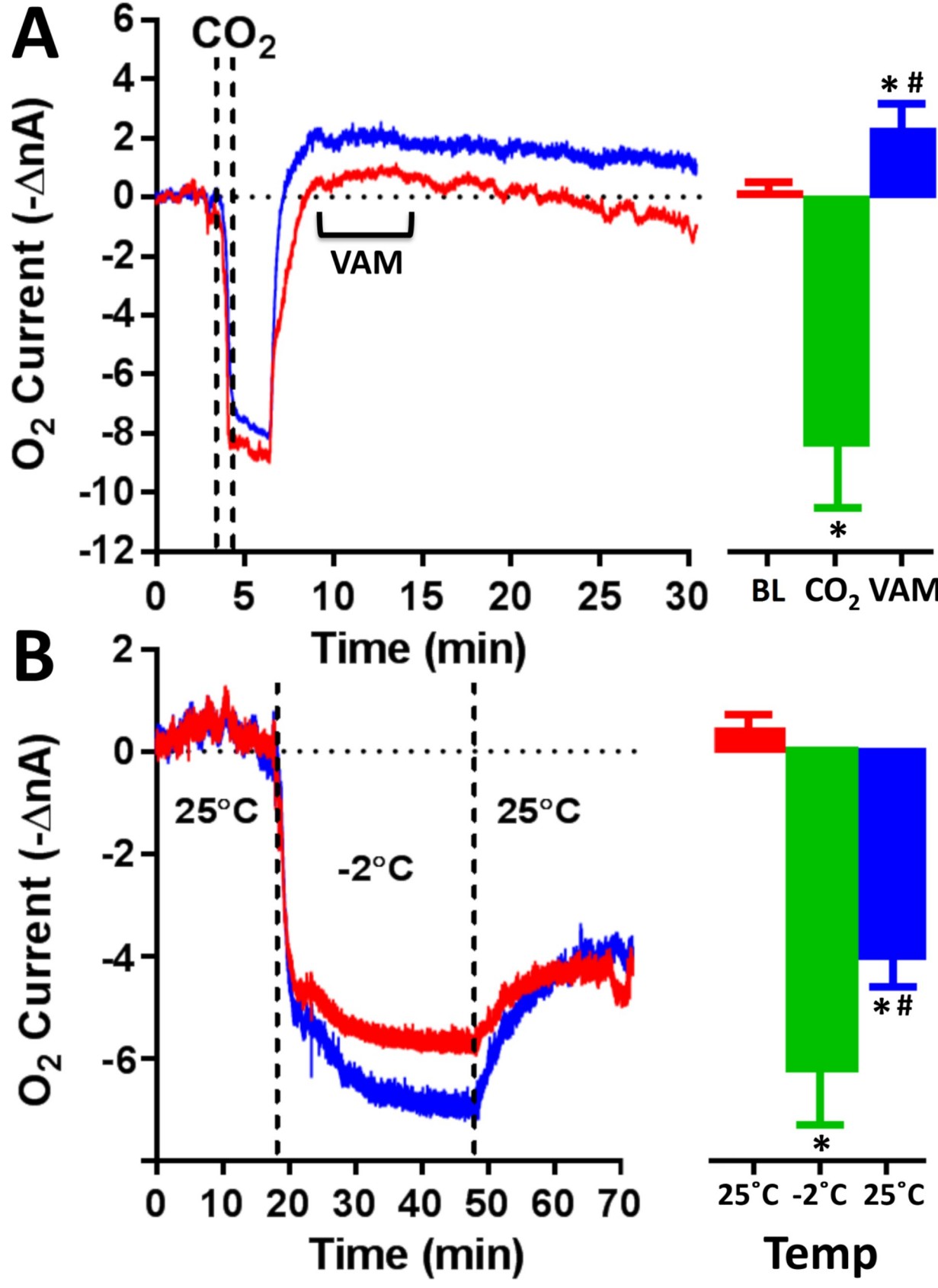

**Fig 5. $CO_2$ and low temperature effects on $O_2$ currents.** Both $CO_2$ (A) and low temperature exposure (B) induced a significant decrease of $O_2$ currents (* = $p < 0.001$ vs baseline). Ventilatory abdominal movements (VAM) were observed after $CO_2$ administration corresponding to a significant increase of the oxygen currents (vs baseline). The exposure of the animals to low temperature resulted in a significant decrease of $O_2$ signals that remained low for few hours. # = $p < 0.05$ vs VAM or -2˚C. BL = baseline.

fixing the feedback resistor (Rf) to 10 MΩ, the I/V converters generated Vout1 e Vout2 signals with an amplification factor of 100 nA/V. The total current range of the amperometric module was 330 nA but only 165 nA could be used for cathodic signals. Finally, the firmware, running on the MCU, performed the voltage-to-current conversions and transmitted them to the USB transceiver. The power consumption was higher than in the past projects [9,10,13,14] and needs further optimizations for extending the battery life, reducing its weight, and integrating the battery inside the device. The choice of the Trinket M0 module was made to allow the simplicity of programming and maintaining the firmware through the use of the innovative CircuitPython interpreter. Until now, the energy savings-libraries for the SAMD21 processor are still missing. A detailed comparison of the significant parameters of the proposed design with similar oxygen-detection telemetry systems is provided in S1 File.

## Oxygen sensors performance

The building of the oxygen sensors used in the present study was based on a previously published procedure [9,10]. The combination of the applied potential with the nitrocellulose coating [24] allowed the interference-free detection of $O_2$. Besides, the deposition of collodion, an high hydrophobic membrane, produced a shielding layer against some poisoning molecules as proteins [24]. The $O_2$ sensor performances were assessed by means of *in-vitro* calibrations for a period of 7 days, confirming previous data on accuracy and precision [9,10], allowing us to trust the oxygen readings made during the 6 days of experiments. Nevertheless, more experiments must be conducted in order to evaluate the effect of long-time exposure to biological environment.

## Oxygen baseline in the extracellular space of CX

Basal oxygen concentrations recorded in the extracellular space of the striatum of freely-moving rats resulted being relatively low (35–50 μM) [9–11] similar to those recorded in the same brain region of mice (~30 μM) [25]. Surprisingly, these observations are in agreement with the results of the present study. These findings appear even more significant in light of the current opinions highlighting the analogies between the central nervous system (CNS) of vertebrates and that of insects [8]. In particular, it has been demonstrated the metabolic coupling between glial cells and neurons might occur in the insect nervous system and may have similarities with that observed in the CNS of vertebrates [8]. The observed oxygen levels turn out to be compatible with the functioning of first-generation enzymatic biosensors [26,27] whereby endogenous molecules as glucose or lactate [23], or exogenous as ethanol [28–30], can be monitored through the implantation of oxidase-based biosensors.

The duration of the experiments did not allow observing significant variations in the basal oxygen levels as can be hypothesized from the results of other authors [31–37]. Prolonged monitoring will be carried out in future research.

## Oxygen changes in the CX after gases, low temperature and anesthetics exposure

In all experiments CX oxygen dynamics resulted in a decline of the local $O_2$ signal, with the sole exception of the animal's exposure to pure oxygen, which produced a significant increase

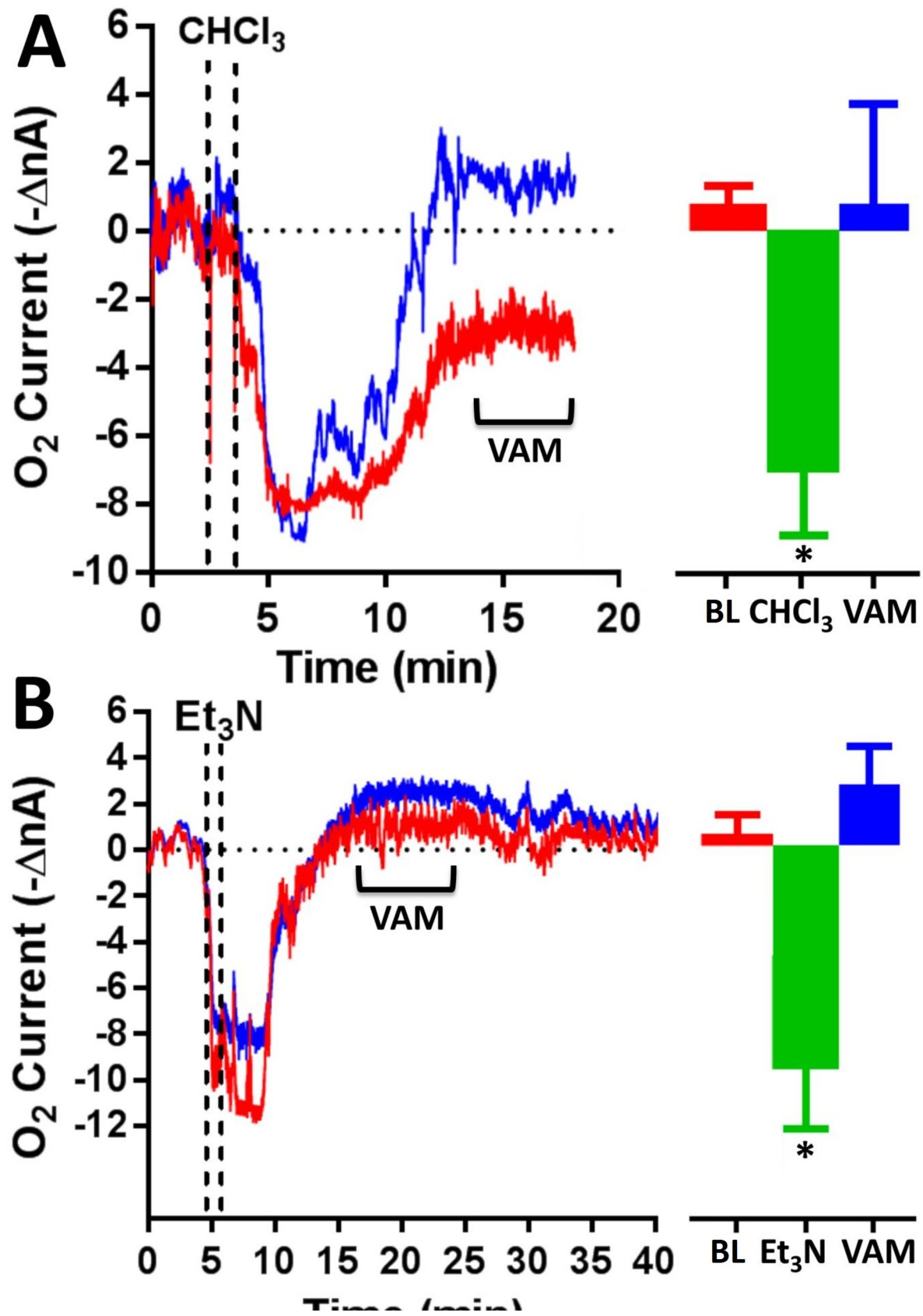

**Fig 6. Chloroform and triethylamine effects on O$_2$ currents.** Chloroform (A) and triethylamine (B) induced a significant decrease of O$_2$ currents (* = p<0.001 vs baseline). VAM were not constantly observed and only for triethylamine post-treatments currents tended to exceed basal values (p>0.05 vs baseline). # = p< 0.05 vs CHCl$_3$ or Et$_3$N. BL = baseline.

of the anodic currents. Direct-administration of N$_2$ (or O$_2$) for 1 min produced a mild hypoxia (or hyperoxia) and resulted in an immediate and significant decrease (or increase) of the oxygen signal until the pure gas was substituted with air. After the administration of pure O$_2$ for 5 min, the CX oxygen levels rapidly exceeded 300 micromoles per liter, returning quickly to basal levels after the replacement of oxygen with air. These evidences confirmed the fast response time of O$_2$ sensors and indicate the rapid penetration of administered gases to the CX (and the fast return to baseline) with a speed comparable with that observed in the striatum of freely-moving rats [9,11]. This convergence of results appears to be surprising especially because the lack of an O$_2$ carrying pigment means the system is intrinsically less efficient than other animal groups that have haemoglobin (or haemocyanin). These findings demonstrate that similar needs, in terms of oxygen supply, are evolutionarily approached (and solved) in different ways.

Both carbon dioxide and low temperature have anesthetic effects in insects [38,39]. The exposure of the cockroach to CO$_2$ induced a transient state of excitation followed by complete immobilization similar to that achieved in small animals [40]. From the neurochemical point of view, during excitement we started to see the drop of oxygen in the CX which remained low for a few minutes. A study performed in *Drosophila melanogaster* [41] showed that during CO$_2$ anaesthesia, the heartbeat stops and oxygen delivery is markedly impaired, generating anoxic conditions. Hypoxia and a disruption of metabolic regulation are reported by MacMillan et al. [39]. In our observations, the subsequent rise in oxygen concentrations (above the baseline) was always associated with VAM (sometimes accompanied by hisses), without awakening from anesthesia. In insects, the appearance of abdominal movements that actively helped breathing was widely documented [31,37,42]. As suggested by Slama [37], "*the actively-regulated breathing of air, based on extracardiac haemocoelic pulsations, has been documented from postembryonic stages of all terrestrial insects, regardless of their overall size and developmental stage*". Acetylcholine, the main neurotransmitter of the autonomic neuroendocrine system in insects, is responsible for the regulation of the extracardiac pulsations, in the same way as the parasympathetic neuroendocrine system regulates the breathing in humans [37].

After exposure to low temperatures, oxygen concentrations dropped sharply and remained low for a few hours. These results are in agreement with Streicher and coworkers [43] who demonstrated a reduction in oxygen consumption and heart rate, both associated with the reduction of temperature in *Gromphadorhina portentosa*. The correlated changes between metabolic rate and cardiovascular function have been shown to be dependent on insect size and temperature [43]. A Similar decrease in O$_2$ consumption was obtained by Matthews and collaborators by applying a Peltier-chilled cold probe to the head of *Nauphoeta cinerea* cockroaches and inactivating their brains [44]. Regarding the validation of the proposed system, the low temperature experiment is important as it is the only one that did not change the gas composition of the surrounding environment, while inducing a significant drop in CX dissolved oxygen.

Chloroform and Triethylamine exert anesthetic effect on *Drosophila melanogaster* [38,45,46]. For both drugs, the onset of anesthesia was inconstant as well as the presence of VAM although the decrease in CX oxygen was always present. The mechanism by which simple molecules can induce anesthesia it is not yet well known but the prevailing hypotheses concern the modification of the lipid membrane fluidity and the interaction with the γ-Aminobutyric acid (GABA) A

ionotropic receptor [47]. Modern anaesthetics are GABA A agonists, induce GABA-mediated neuronal hyperpolarization and, by causing a dose-dependent suppression of cerebral metabolism, reduce neural demand of oxygen and glucose [48]. Immunohistochemical staining confirmed the presence of GABA A receptors in insects [49]. Moreover, GABA immunostaining in the CX of a praying mantis (*Hierodula membrana*) and three cockroach species (*Rhyparobia maderae*, *Blaberus discoidalis* and *Periplaneta americana*) showed three apparently homologous systems of neurons [50]. The latter observations indicate that patterns of GABA innervation in the CX are remarkably conserved throughout dicondylian insects [50].

## Conclusions

In this study we present a novel telemetric system for the real-time detection of dissolved oxygen changes in the central complex of *Gromphadorhina portentosa*. The telemetric device, coupled with two implantable electrochemical sensors, is able to send $O_2$ reduction signals to a transceiver moduled soldered to a USB dongle (notebook). The proposed system revealed good features in terms of potentiostat stability, nanocurrent conversion and consistent neurochemical data transmission. For the first time, the *in-vivo* results highlighted $O_2$ dynamics in the CX under different experimental conditions. For these reasons, the proposed system can constitute a new experimental model for the exploration of central complex neurochemistry and it can also work with oxidizing sensors and amperometric biosensors.

The deep homology between the central complex of arthropods and the basal ganglia of vertebrates [51] may be useful to broaden the knowledge of the basal ganglia neurophysiology and certain basal ganglia-related disorders such as Parkinson's and Huntington's diseases.

Finally, neurosensor monitoring can provide bioenergetics feedback to new neural interfaces developed for controlling insects' behavior [52] and to clarify unknown neurochemical aspects of the movement control in the central complex [53].

## Supporting information

**S1 File. Detailed description of the telemetry system (hardware, firmware and software) and the stereotaxic adapter.**
(DOCX)

**S1 Movie. Short footage showing a freely-running cockroach connected to the telemetry system.**
(MP4)

## Acknowledgments

The authors acknowledge Arizona Microchip for the free MCP6044 samples.

## Author Contributions

**Conceptualization:** Pier Andrea Serra.

**Data curation:** Pier Andrea Serra, Patrizia Monti.

**Funding acquisition:** Gaia Rocchitta.

**Investigation:** Paola Arrigo, Andrea Bacciu, Daniele Zuncheddu, Riccardo Deliperi, Diego Antón Viana.

**Methodology:** Maria Vittoria Varoni, Maria Alessandra Sotgiu, Pasquale Bandiera.

**Resources:** Maria Vittoria Varoni, Maria Alessandra Sotgiu, Pasquale Bandiera.

**Writing – original draft:** Pier Andrea Serra, Gaia Rocchitta.

**Writing – review & editing:** Pier Andrea Serra, Gaia Rocchitta.

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
