## [Decision Letter · Decision Letter 0]

16 Oct 2019

PONE-D-19-22207

Real-time Telemetry Monitoring of Oxygen in the Central Complex of freely-walking Gromphadorhina portentosa

PLOS ONE

Dear Dr. Serra,

Thank you for submitting your manuscript to PLOS ONE. After careful consideration, we feel that it has merit but does not fully meet PLOS ONE’s publication criteria as it currently stands. Therefore, we invite you to submit a revised version of the manuscript that addresses the points raised during the review process.

We would appreciate receiving your revised manuscript by Nov 30 2019 11:59PM. To enhance the reproducibility of your results, we recommend that if applicable you deposit your laboratory protocols in protocols.io, where a protocol can be assigned its own identifier (DOI) such that it can be cited independently in the future. For instructions see: http://journals.plos.org/plosone/s/submission-guidelines#loc-laboratory-protocols

We look forward to receiving your revised manuscript.

Kind regards,

Kalisadhan Mukherjee

Academic Editor

PLOS ONE

Journal Requirements:

3. Please ensure that you refer to Figure 2 in your text as, if accepted, production will need this reference to link the reader to the figure.

Additional Editor Comments:

In the manuscript entitled “Real-time Telemetry Monitoring of Oxygen in the Central Complex of freely-walking Gromphadorhina portentosa” authors presented a telemetric system for detection of dissolved oxygen changes in the central complex (CX) of Gromphadorhina portentosa. Similar studies are also carried out by the author’s group and reported in the literature (Anal. Chem. 2013, 85, 10282−10288). Please mention the new findings or novel ideas that are described in the present article as compared to the literature published by the author’s group in the same area.

In order to monitor the changes of extracellular oxygen levels, amperometric sensors have been employed. Did the authors measure in vitro the oxygen detection performance of these sensors? Comment on it.

Describe the methodologies adopted to calibrate the sensors.

Reviewers' comments:

Reviewer's Responses to Questions

**Comments to the Author**

1. Is the manuscript technically sound, and do the data support the conclusions?

Reviewer #1: Yes

2. Has the statistical analysis been performed appropriately and rigorously? 

Reviewer #1: Yes

3. Have the authors made all data underlying the findings in their manuscript fully available?

Reviewer #1: Yes

4. Is the manuscript presented in an intelligible fashion and written in standard English?

Reviewer #1: Yes

5. Review Comments to the Author

Reviewer #1: This paper is well written and presented. The author studied a novel telemetric system by detecting dissolved oxygen changes in the central complex of Gromphadorhina portentosa. Two electrochemical sensors have been used to monitor the reduction of oxygen. However, the author can include a comparison table with a few recent articles to show the advantages of oxygen monitoring in terms of stability, power consumption, and other parameters related to the oxygen detection in the telemetric system.

6. PLOS authors have the option to publish the peer review history of their article (what does this mean?). If published, this will include your full peer review and any attached files.

Reviewer #1: No

---

## [Author Response · Author response to Decision Letter 0]

22 Oct 2019

Point-by-point response to the Academic Editor's comments

I would like to thank you for the time you spent for editing the manuscript entitled “Real-time Telemetry Monitoring of Oxygen in the Central Complex of freely-walking Gromphadorhina portentosa” that I submitted to be considered for publication in the PLOS ONE Journal. I also thank you for the comments and the suggestions that will help to improve the impact of the study. 

1) "Please mention the new findings or novel ideas that are described in the present article as compared to the literature published by the author’s group in the same area".

As highlighted in the conclusion paragraph of the manuscript, “in this study we present a novel telemetric system for the real-time detection of dissolved oxygen changes in the central complex of Gromphadorhina portentosa”. The proposed telemetry system is innovative in terms of component used for hardware design and firmware programming. The open source design allows to easily replicate the device and to adapt it to future applications such as, for example, the use of first generation enzyme-based amperometric biosensors. However, the most innovative idea consists in monitoring the brain oxygen itself in the central complex of an insect large enough to allow the use of neurosurgical techniques, already developed for vertebrates. This has been made possible thanks to the development and adaptation of different techniques and procedures developed over the years on vertebrates, rodents in particular. For these reasons, the proposed system can constitute a new experimental model for the exploration of central complex neurochemistry. Indeed, “neurosensor monitoring can provide bioenergetics feedback to new neural interfaces developed for controlling insect behavior and to clarify unknown neurochemical aspects of the movement control in the central complex”. 

Following the observation made by the Academic Editor, we searched for the possible analogies between the central complex and structures of the central nervous system of vertebrates previously studied by our research group. We found a very interesting review entitled “Deep Homology of Arthropod Central Complex and Vertebrate Basal Ganglia” by Nicholas J. Strausfeld and Frank Hirth published in Science Journal and we added the following paragraph to the Conclusion section:

“The deep homology between the central complex of arthropods and the basal ganglia of vertebrates [51] may be useful to broaden the knowledge of the basal ganglia neurophysiology and certain basal ganglia-related disorders such as Parkinson's and Huntington's diseases”.

2) "In order to monitor the changes of extracellular oxygen levels, amperometric sensors have been employed. Did the authors measure in vitro the oxygen detection performance of these sensors? Comment on it".

An accurate in-vitro characterization of the oxygen sensors was done in a previous study (Bazzu G, Puggioni GM, Dedola S, Calia G, Rocchitta G, Migheli R, et al. Real-time monitoring of brain tissue oxygen using a miniaturized biotelemetric device implanted in freely moving rats. Anal Chem. 2009;81: 2235–2241. doi:10.1021/ac802390f).

Cyclic voltammetry (CV) has been used for the identification of the potential threshold for oxygen reduction on epoxy carbon surface (-350 mV vs Ag/AgCl pseudo-reference electrode) while constant potential amperometry (CPA) was used for the precise quantification of oxygen concentrations (-400 mV vs Ag/AgCl). The pre-implantation in-vitro calibrations have been used for the quantification of central complex extracellular oxygen during in-vivo experiments. As discussed in the paragraph “Oxygen sensors performance”, “the O2 sensor performances were assessed by means of in-vitro calibrations for a period of 7 days, confirming previous data on accuracy and precision […], allowing us to trust the oxygen readings made during the 6 days of experiments”.

Unfortunately, due to the reduced length of the sensors, we could not extract them intact from the animals' heads to perform a post-implantation calibration. However, the sensors' responses to the exposure of animals to pure oxygen and nitrogen gases confirmed the sensitivity of the sensors at least during the first week of experiments. Finally, as discussed in the manuscript, “more experiments must be conducted in order to evaluate the effect of long-time exposure to biological environment”.

3) "Describe the methodologies adopted to calibrate the sensors ".

In the paragraph entitled “Preparation and calibration of oxygen sensors” we described the methodologies adopted to calibrate the sensors. We wrote: “A precise calibration was carried out at low concentrations of O2 (Fig 1, upper-left inset) after having connected the microsensors to the telemetric device (see the dedicated paragraph) and adding up, to a 10 milliliters of N2-purged PBS, defined volumes of a 100% O2 solution”.

In order to better describe the procedure without weighing down the manuscript, we just added the increasing volumes of 100% O2 solution used for performing the calibration (+200, +204, +208, +212, and +216 μL). More details on calibration procedures can be found in the above-cited study published in a journal of pure analytical chemistry (Bazzu et al., 2009).

Final checks

We carefully checked the journal style requirements and the data sharing requirements; in particular all occurrences of the phrase “data not shown” in the manuscript have been removed and/or replaced with references to our previous studies. Finally, the reference to Figure 2 has been included in the text of the main manuscript.

Point-by-point response to the Reviewer's comments (Reviewer #1)

I would like to thank the Reviewer for the time He/She spent in reviewing the manuscript and the suggestions that will help to improve the impact of the study.

Reviewer #1 comments and suggestions:

1) "[...] the author can include a comparison table with a few recent articles to show the advantages of oxygen monitoring in terms of stability, power consumption, and other parameters related to the oxygen detection in the telemetric system".

Accepting the reviewer's suggestion we have identified (in recent literature) two telemetric oxygen detection systems similar to the one proposed in the present study and we compared them using a table. The first system is the one developed by our work group in 2009 (Bazzu et al., 2009) while the second was developed by Russel and coworkers in 2012. In order to provide the required details without weighing down the main manuscript, a paragraph entitled “Comparison among oxygen-detection telemetry systems” has been added to the Supporting Information.

---

## [Editor Report · Decision Letter 1]

25 Oct 2019

Real-time Telemetry Monitoring of Oxygen in the Central Complex of freely-walking Gromphadorhina portentosa

PONE-D-19-22207R1

Dear Dr. Serra,

We are pleased to inform you that your manuscript has been judged scientifically suitable for publication and will be formally accepted for publication once it complies with all outstanding technical requirements.

With kind regards,

Kalisadhan Mukherjee

Academic Editor

PLOS ONE

Additional Editor Comments (optional):

Authors have revised carefully the manuscript in accordance to the comments made by the reviewer and academic Editor. The manuscript now can be accepted for publication in PlosOne.
---

## [Editor Report · Acceptance letter]

30 Oct 2019

PONE-D-19-22207R1 

Real-time Telemetry Monitoring of Oxygen in the Central Complex of freely-walking *Gromphadorhina portentosa*

Dear Dr. Serra:

I am pleased to inform you that your manuscript has been deemed suitable for publication in PLOS ONE. Congratulations! Your manuscript is now with our production department. 

With kind regards,

on behalf of

Dr. Kalisadhan Mukherjee 

Academic Editor

PLOS ONE